# Description of Gut Mycobiota Composition and Diversity of Caprinae Animals

Qing-Bo Lv,[a,b,c] Jin-Xin Meng,[a,c] He Ma,[a,c] Rui Liu,[a,c] Ya Qin,[a,c,d] Yi-Feng Qin,[a,c,d] Hong-Li Geng,[a,c,d] Hong-Bo Ni,[a,c] Xiao-Xuan Zhang[a]

[a]College of Veterinary Medicine, Qingdao Agricultural University, Qingdao, Shandong Province, People's Republic of China
[b]College of Veterinary Medicine, Jilin University, Changchun, Jilin Province, People's Republic of China
[c]Key Laboratory of Bovine Disease Control in Northeast China, Ministry of Agriculture and Rural affairs of the People's Republic of China, Heilongjiang Provincial Key Laboratory of Prevention and Control of Bovine Diseases, College of Animal Science and Veterinary Medicine, Heilongjiang Bayi Agricultural University, Daqing, China
[d]College of Animal Science and Technology, Jilin Agricultural University, Changchun, Jilin Province, People's Republic of China

Qing-Bo Lv and Jin-Xin Meng contributed equally to this article. Author order was determined alphabetically.

**ABSTRACT** The fungal community, also known as mycobiota, plays pivotal roles in host nutrition and metabolism and has potential to cause disease. However, knowledge of the gut fungal structure in Caprinae is quite limited. In this study, the composition and diversity of the gut mycobiota of Caprinae animals from different geographical locations (Anhui, Jilin, Guangxi, Shandong, Shanxi, and Tibet) were comprehensively characterized by analyzing the internal transcribed spacer 2 (ITS-2) sequences of the fungal community. The results showed that *Ascomycota* and *Basidiomycota* were the dominant phyla, which, respectively, accounted for 90.86 to 95.27% and 2.58 to 7.62% of sequences in samples from each region. Nonetheless, the structure of the gut mycobiota was largely different in Caprinae animals in the different provinces. Therein, *Sporormiaceae* and *Thelebolaceae* were the dominant fungal families in the samples from Tibet, whereas their abundance was generally low in other regions. The intestinal diversity of individuals from Guangxi was higher than that in other regions. In addition, there were 114 differential genera among all regions. Finally, the co-occurrence network revealed 285 significant correlations in cross-family pairs in the guts of Caprinae animals, which contained 149 positive and 136 negative relationships, with 96 bacterial and 86 fungal participants at the family level. This study has improved the understanding of the mycobiota of ruminants and provided support for the improvement in animal health and productivity.

**IMPORTANCE** In this study, we elucidated and analyzed the structure of the gut mycobiota of Caprinae animals from different regions. This study revealed differences in the structure of the gut mycobiota among Caprinae animals from different geographical environments. Based on previous findings, correlations between fungal and bacterial communities were analyzed. This study adds to previous research that has expanded the present understanding of the gut microbiome of Caprinae animals.

**KEYWORDS** Caprinae, gut mycobiota, ITS amplicon sequencing, geographic variation

Address correspondence to Xiao-Xuan Zhang, zhangxiaoxuan1988@126.com.

The authors declare no conflict of interest.

The gastrointestinal tract (GIT) microbiota of humans and animals is one of the most complicated microbial ecosystems known. It includes bacteria, fungi, archaea, protozoa, and viruses and is vital in the regulation of biological processes associated with nutrient absorption and homeostatic maintenance (1, 2). The gut microbiota of humans, murine species, livestock, and avian species have been intensively explored in recent years thanks to the advent of next-generation sequencing (3–6). The fungal communities in the gut have received little attention over the past few decades compared to the bacterial ones. Fungi are traditionally studied via culture-dependent methods, which limits the in-depth understanding of the gut mycobiota (2, 7, 8).

In humans, the mycobiota is estimated to account for approximately 0.1% of the total microbes in the gut (1, 9). At the level of phyla, most studies thus far have suggested that *Ascomycota*, *Basidiomycota*, and *Zygomycota* are the predominant phyla in the gut (2, 10). Fewer data are available for animal mycobiota in comparison with that of humans. Oligonucleotide fingerprinting of rRNA gene analysis and high-throughput sequencing have provided evidence that the murine intestine harbors a diverse array of fungi. *Ascomycota* and *Basidiomycota* have been identified as the major phyla, with *Candida*, *Saccharomyces*, *Trichosporon*, *Aspergillus*, *Penicillium*, *Wickerhamomyces*, *Cladosporium*, and *Fusarium* being among the most abundant genera (11). In the piglet mycobiota, *Ascomycota* and *Basidiomycota* are the dominant phyla. The mycobiota in piglets change significantly from before to after weaning, a process that involves a transition from a predominance of *Cladosporiaceae* to one of *Saccharomycetaceae*, as well as the introduction of *Dipodascaceae* and *Aspergillaceae* (12, 13). In our previous work (not yet published), we found that the mycobiota in the broiler gut was mainly composed of *Ascomycota* and *Basidiomycota*, which were increased significantly in the middle and later stages of life, and *Candida*, *Trichosporon*, and *Aspergillus* were the dominant genera. Moreover, a recent study regarding the chicken mycobiota suggested that *Gibberella* was also of the most abundant genera (14). In ruminants, research on mycobiota has primarily focused on fungi in the rumen, with special attention given to anaerobic fungi belonging to the phylum *Neocallimastigomycota* (15, 16). Amplicon sequencing in Ujimqin sheep revealed that the large intestine was largely colonized by *Dothideomycetes* and *Leotiomycetes*, while the stomach and small intestine were preferentially colonized by *Neocallimastigomycetes* and *Sordariomycetes*, respectively. Meanwhile, *Cryptococcus* was significantly enriched in the small intestine, and *Sporormiaceae* was significantly dominant in the large intestine (17). However, a detailed in-depth analysis of the gut mycobiota of Caprinae animals in different geographical environments is absent.

Caprinae animals (e.g., goats, sheep, and antelopes) display species polymorphism. They are well adapted to different elevations, geographies, climates, and feeding environments. Temperature, vegetation, and water in these habitats significantly affect the fiber digestion and nutrient absorption strategies of ruminants (18). In this study, we analyzed gut fungal communities using the Illumina sequencing of amplicon libraries that targeted fungi to thoroughly decipher the composition and diversity of the gut mycobiota of Caprinae species in different provinces of China. Moreover, combined with the metagenomic sequencing data set from our previous study, a co-occurrence network of bacteria and fungi was constructed to explore the cross talk between the gut bacterial microbiota and mycobiota, as well as their contributions to gut homeostasis and host health. This study was the first to compare the gut mycobiota of Caprinae animals in different geographical environments. These data are expected to support further studies investigating the relationship between the gut mycobiota and host health.

## RESULTS

**The mycobiota diversity in the guts of Caprinae animals.** To analyze the mycobiota in Caprinae animals, the fungal internal transcribed spacer 2 (ITS-2) region was amplified and sequenced for 28 fecal samples. A total of 1,118,317 high-quality reads were obtained via the QIIME2 processing and filtering pipeline, resulting in a total of 1,848 amplicon sequence variants (ASVs) for all samples. Among them, 1,768 (95.67%), 1,676 (90.69%), and 1,539 (83.28%) ASVs were identified according to order, family, and genus, respectively. In addition, the rarefaction curves tended to reach a saturation plateau, indicating that the combined mycobiota of the 28 samples was large enough to estimate the diversity of the fungal community (Fig. S1B in the supplemental material).

The Shannon index and the number of ASVs were calculated to measure the $\alpha$-diversity in the mycobiota of Caprinae animals in different provinces (Fig. 1A). Compared to other regions, Guangxi province had the highest number of ASVs and the highest Shannon index. However, the fungal diversities were not significantly different from other provinces with respect to Guangxi province. The principal coordinate analysis (PCoA) showed clear clustering of the gut mycobiota for each region; therein, the cluster of samples among

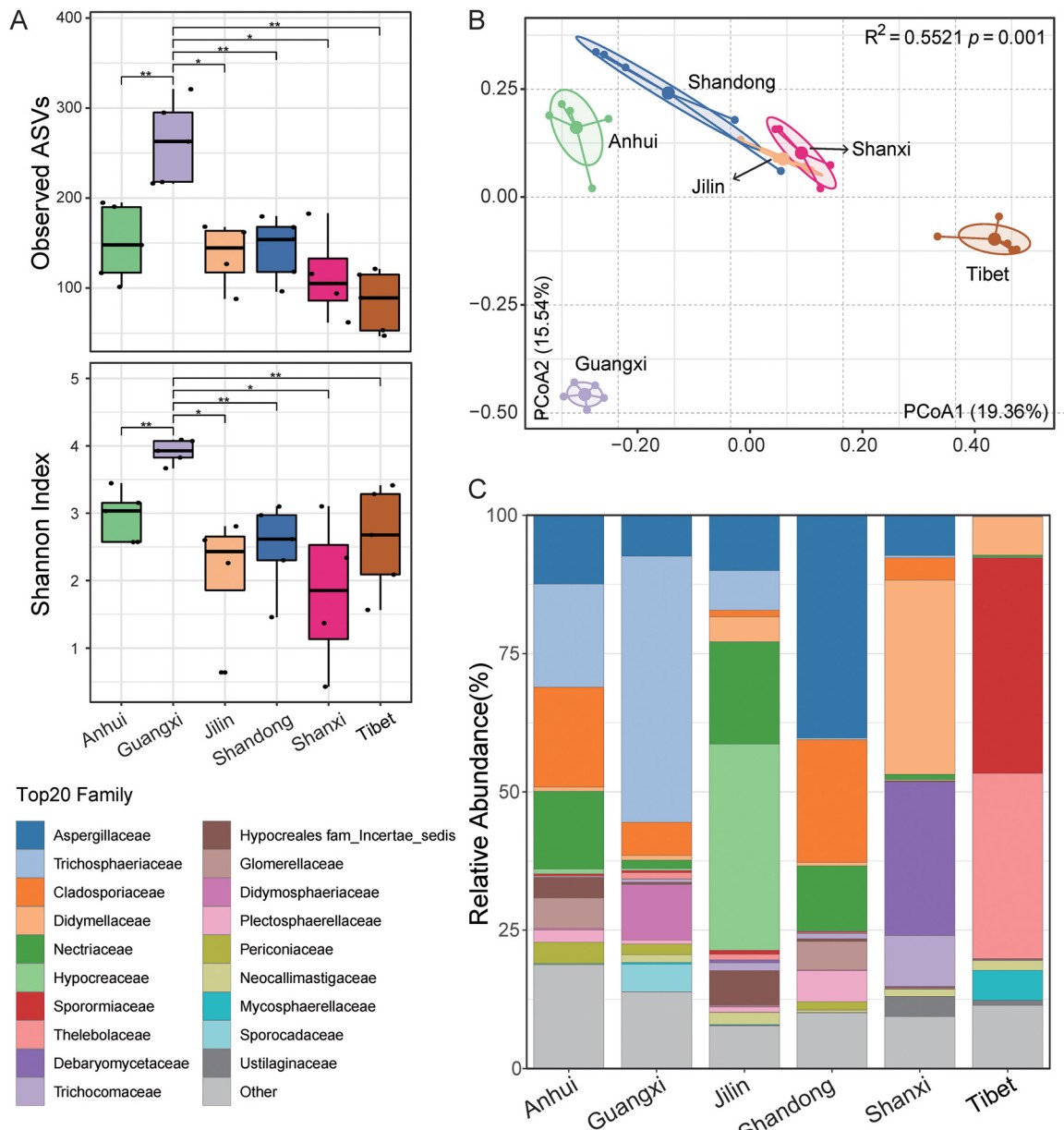

**FIG 1** Composition and diversity of gut mycobiota in animals from difference provinces. (A) Community composition of the gut mycobiota in animals from six provinces at the family level. (B) α-Diversity of gut mycobiome, including observed amplicon sequence variants (ASVs) and Shannon index. Wilcoxon rank sum tests were employed to analyze differences between groups. Statistical significance: *, $P < 0.05$; **, $P < 0.01$. (C) β-Diversity shows the structural differences in the gut communities of fungi. Principal coordinate analysis (PCoA) and permutation multiple variance analysis were used to reveal differences in gut fungal communities.

Shandong, Jilin, and Shanxi provinces had a closer distance. Further investigation using permutation multiple variance analysis (PERMANOVA) confirmed that the mycobiota structure of the guts of Caprinae animals exhibited significant differences among the different provinces ($R^2 = 0.5521$, $P = 0.001$, Fig. 1B).

**The mycobiota structure in the guts of Caprinae animals.** Taxonomic distributions of fungal ASVs at different classification levels were assessed to uncover the mycobiota structure. *Ascomycota* and *Basidiomycota* were found to be the dominant phyla, accounting for 90.86 to 95.27% and 2.58 to 7.62% of the sequences in each geographic region, respectively (Table S4). In contrast, *Chytridiomycota* and *Mortierellomycota* were found only in the individuals from Shanxi and Anhui provinces, respectively (Fig. S1C). In addition, significant differences ($P < 0.05$) in the abundance between *Mucoromycota* and *Neocallimastigomycota* were

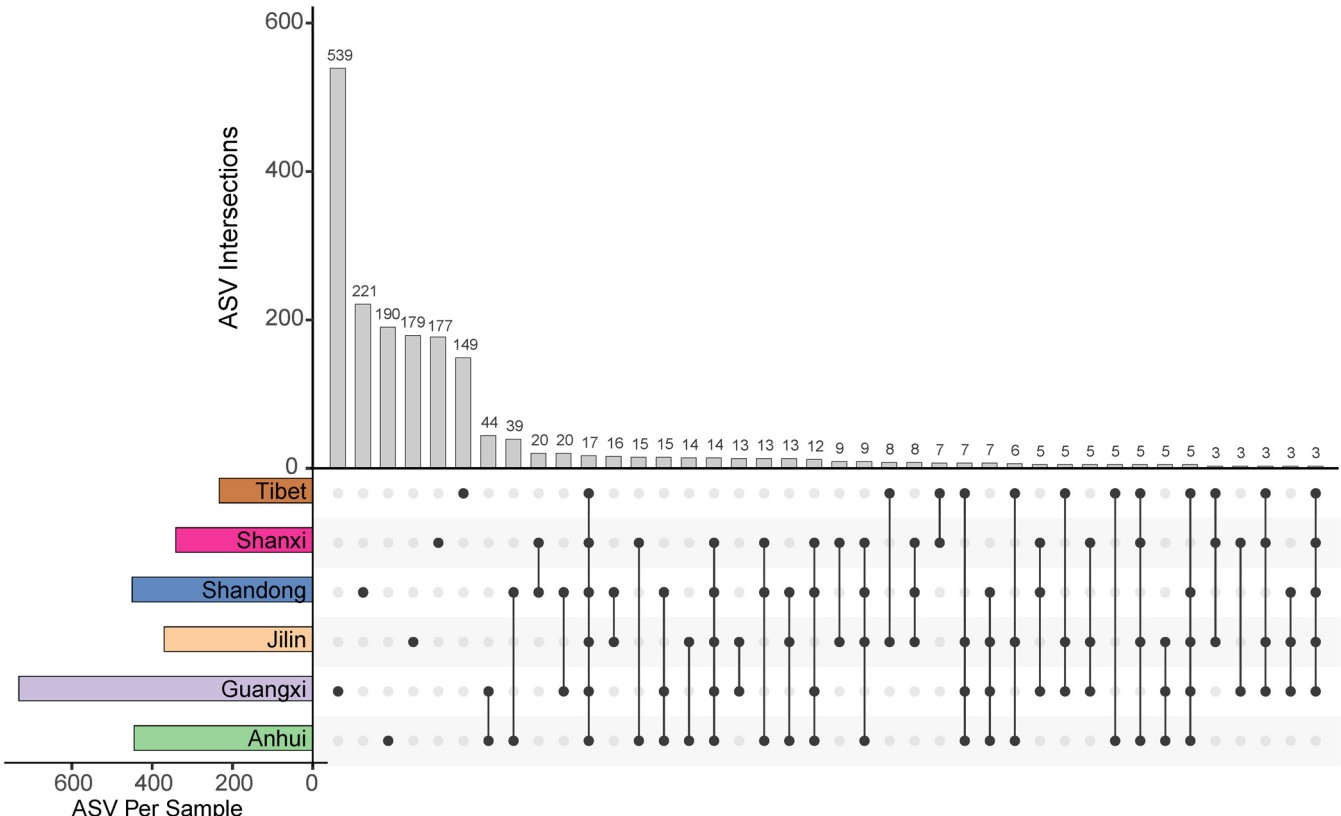

**FIG 2** Distribution of ASVs across the guts of Caprinae animals in different provinces.

detected in different provinces. At the family level, Caprinae animals in different regions showed obvious variations in their gut fungal structure. *Sporormiaceae* and *Thelebolaceae* were the dominant families in Tibet, where their abundances was significantly higher ($P < 0.05$) than in other regions. *Didymellaceae* and *Debaryomycetaceae* were the dominant families in Shanxi province. Moreover, the abundance of *Trichosphaeriaceae* in the samples from Guangxi province was nearly 50%, which was significantly higher ($P < 0.05$) than that in other regions (Fig. 1C; Fig. S2). At the genus level, different dominant fungal genera were detected in the samples from six provinces. For example, the genera *Thelebolus*, *Sporormiella*, *Mycocentrospora*, and *Mycocentrospora* were dominant in the Tibetan samples. Interestingly, the abundances of genera *Thelebolus* and *Sporormiella* in samples from other regions were low (relative abundance $< 1\%$). It is noteworthy that a high abundance of *Debaryomyces* (43.56%) was found in samples from Shanxi province, but this genus was not found in other regions (Fig. S1; Table S4).

**Common, unique, and core fungal ASVs in the gut mycobiota of Caprinae animals.** We further investigated the distribution of common and unique ASVs in the gut mycobiota of Caprinae animals in different provinces. Guangxi province had the highest number of specific ASVs (539/1,848, 29.17%), while Tibet had the lowest number (149/1,848, 8.06%) (Fig. 2). Most of the unique ASVs ($>90\%$) were classified as *Ascomycota* and *Basidiomycota*. The concept of the "core microbiota" is used to identify and describe key microorganisms that were stable and permanent in a microbial community (19). A total of 17 ASVs were shared among the samples from all regions, and they were all assigned to *Ascomycota* (Fig. 2). Of these, 15 ASVs could be classified at the genus level, including *Albifimbria*, *Aspergillus*, *Chordomyces*, *Cladosporium*, *Colletotrichum*, *Fusarium*, *Nigrospora*, *Penicillium*, *Periconia*, *Sporormiella*, *Stephanonectria*, and *Trichoderma*. These core fungi may be important members of the gut microbial structure in Caprinae animals.

To provide more information about the gut fungal communities of Caprinae

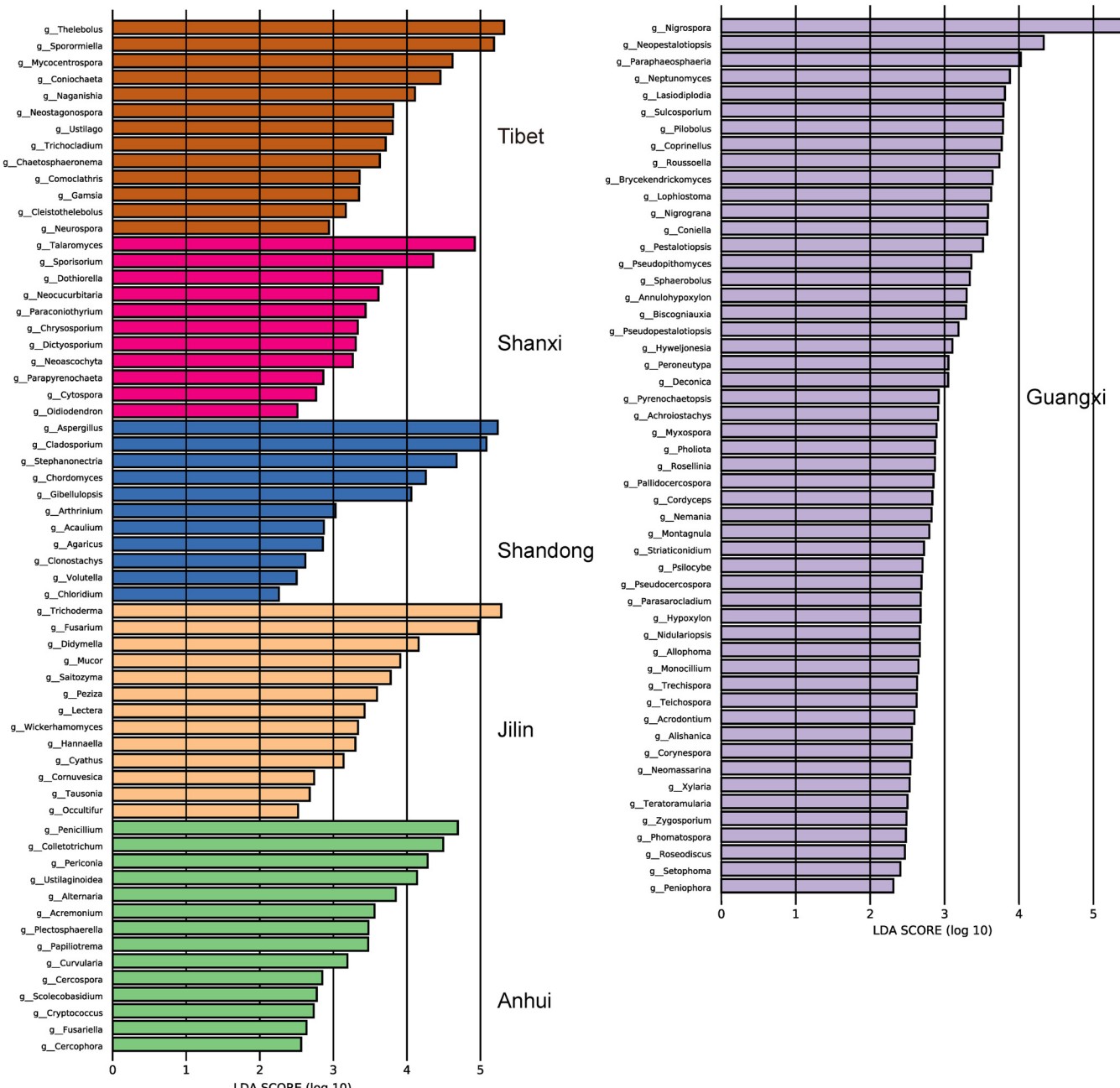

**FIG 3** Taxa with significantly different abundances at the genus level are shown. The higher the linear discriminant analysis (LDA) score, the greater the effect of the taxa abundance on the differences among the guts of Caprinae animals in six provinces.

animals in different provinces, linear discriminant analysis (LDA) of the effect size (LEfSe) was employed to identify differentially abundant genera among six provinces with LDA scores higher than 2.0. The LEfSe analysis of the gut fungal communities in Caprinae animals showed that there were 114 distinctly abundant genera across all provinces (Fig. 3; Table S5). These fungal genera belonged principally to *Ascomycota* and *Basidiomycota* phyla. It is worth noting that individuals from Guangxi province had the highest number of differential fungi, suggesting that animals from this region have a unique composition of gut mycobiota.

**Bacterial microbiota interactions with the mycobiota in the different provinces.** The bacterial abundance data came from a previously published study based on metagenomic sequencing (20). Potential interactions among families found in the gut microbiota (bacteria and archaea) and mycobiota were determined using Spearman

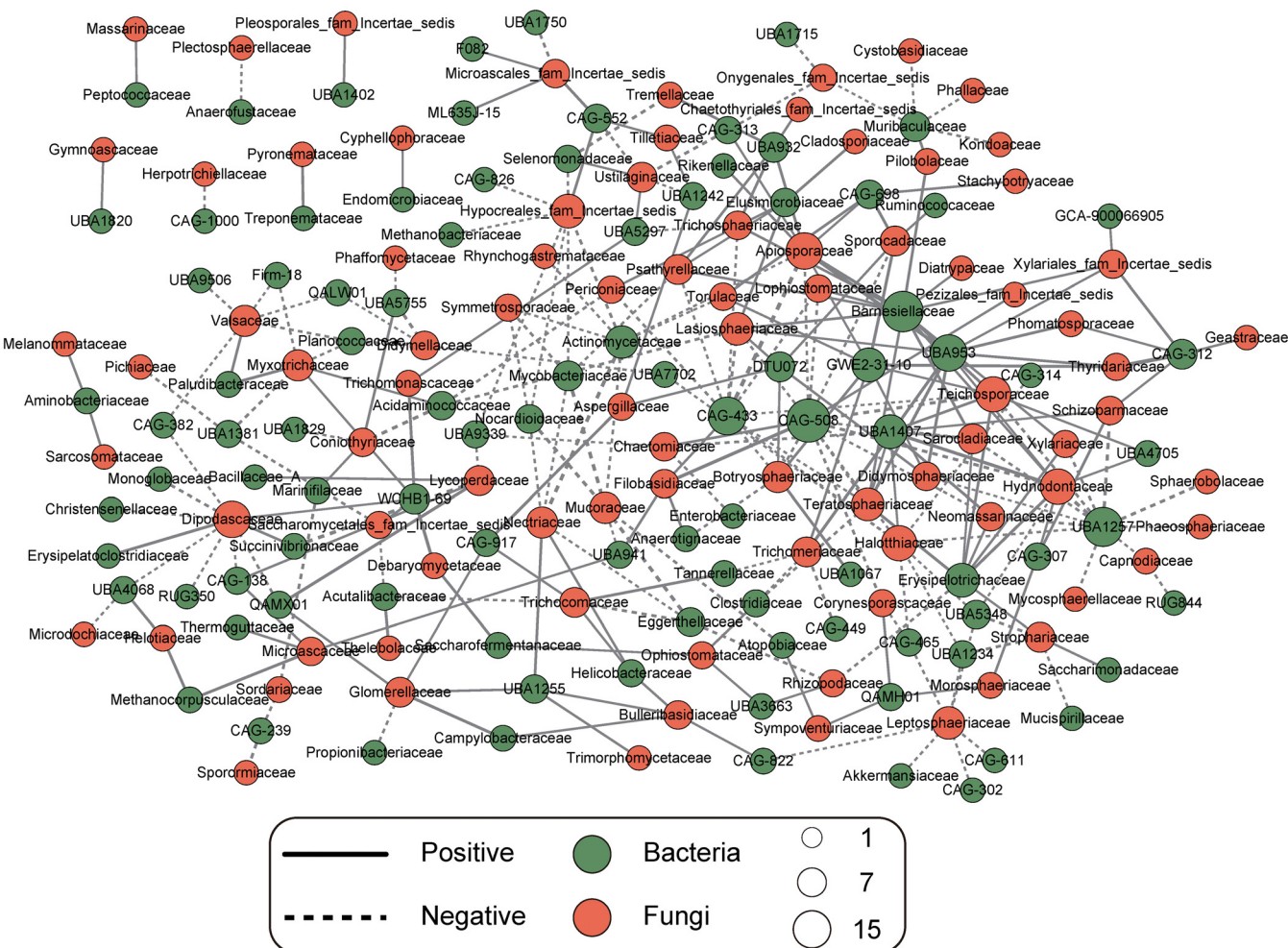

**FIG 4** Network based on Spearman correlation coefficients ($P < 0.05$) showing connectedness between bacterial microbiota and mycobiota at the family level in the guts of Caprinae animals in different provinces. Solid lines represent significantly (>0.5) positive linear relationships; dashed lines represent (less than −0.5) negative linear relationships. The width of the line is proportional to the strength of the relationship. Green nodes represent bacteria and orange nodes represent fungi, and the size of each node is proportional to the number of connections.

correlations and corresponding network analysis, revealing 285 significant correlations in cross-family pairs in the gut. The co-occurrence network had 96 bacterial and 86 fungal participants at the family level, and it contained 149 positive and 136 negative relationships (Fig. 4). Therein, the fungal family *Thyridariaceae* and the bacterial family CAG-312 showed the strongest positive correlation, while the fungal family *Apiosporaceae* and the bacterial family CAG-433 had the strongest negative correlation. In addition to this, the fungal families *Apiosporaceae* and *Dipodascaceae* had the most interactions with bacteria, and the bacterial family CAG-508 was the most frequently observed to interact with fungi (Table S6).

## DISCUSSION

The accumulated evidence clearly indicates that the gut mycobiota has an indisputable role in host homeostasis and disease development, despite constituting only a small proportion of the gut (2, 21). However, studies examining the mycobiota in the gut are limited. To the best of our knowledge, few studies have been performed to explore the mycobiota characteristics in the guts of Caprinae animals from different geographical environments. The objective of this study was to explore the differences in gut fungal diversity and composition patterns in Caprinae animals from China and to perform a further evaluation of the interactions between the microbiota and

mycobiota. Our data showed that the gut mycobiota structure and diversity of Caprinae animals from different regions were significantly different. The co-occurrence networks demonstrated complex interactions between gut microbiota and mycobiota. Thus, it is suggested that the geographical environment exerts a significant influence on the composition and diversity of the gut mycobiota in Caprinae animals.

A previous study showed that the soil in the Tibetan plateau was dominated by classes *Dothideomycetes* and *Sordariomycetes* (22). Thus, the high abundance of classes *Dothideomycetes* and *Sordariomycetes* in the gut of *P. hodgsonii* may be associated with their habitats and feeding habits, such as high probability of contact with grassland soil during grazing. The Caprinae animals from Guangxi showed high gut mycobiota diversity. We speculate that there are more abundant edible plants in this region for Caprinae animals, as regions in Guangxi province have higher temperature and humidity. Bahram et al. (23) also showed that plant and plant-derived C may be a strong factor in shaping fungal community composition. Thus, diversity may be induced by geographical factors in different regions. Notably, a metagenomic study showed that the dominant fungi in the gut were a subset of those found in external host habitats, which were in turn a subset of those found in environmental habitats (23). The gut mycobiota may be greatly affected by the environmental factors, such as altitude, temperature, and moisture, in different regions. Nonetheless, due to the limitations of the current study, it is still necessary to clarify whether a given fungal genus is an original inhabitant or an alien in the gut. Moreover, the mechanism inducing the formation of different compositions of gut mycobiota in Caprinae animals from different provinces under geographical environmental stress needs further investigation.

We found that some fungi were prevalent in all samples. This group included fungi commonly found in soil, such as *Aspergillus*, *Periconia*, *Nigrospora*, and *Trichoderma* (24 to 27), and some plant fungi, such as *Albifimbria*, *Colletotrichum*, *Cladosporium*, and *Fusarium* (28–31). There were also fungi that have been reported to be prevalent in herbivores, such as *Sporormiella* (32). These core fungi may play an important role in the intestinal ecology of Caprinae animals. In addition, samples from Guangxi showed a unique fungal community structure, with high abundances of *Nigrospora* and *Neopestalotiopsis*. *Nigrospora* is a common soil fungus, and many species of this genus are considered to be plant pathogens (33). *Neopestalotiopsis* is a fungal pathogen that affects strawberries and many other crops (34). In addition, we also found a high abundance of *Debaryomyces* in the Shanxi samples. This is an ascomycetous yeast commonly found in soil, water, and plants (35). Its member species *Debaryomyces hansenii* has been extensively studied for its salt tolerance and potential probiotic effects (36). This suggests that the fungus is likely to be widespread in the local environment. The accumulation of these fungi may reflect the dietary structure of Caprinae animals in the area, so differences in the dietary composition may be the main reason for the variation in gut mycobiota in animals from different areas.

Complex interactions between bacteria and fungi also occur in the gut. Some significant correlations were found between bacterial and fungal families in the guts of Caprinae animals, thus suggesting potential relationships between the bacterial microbiota and mycobiota. Negative interactions between species may be induced by competition for limited resources or spaces, causing them to attack each other (37). Thus, it seems plausible that strong competitive relationships between fungi and bacteria in the gut suppress certain types of fungal growth. Moreover, some fungi and bacteria are involved in mutualistic relationships that help to maintain the stability of the gut microbiome (37). Co-occurrence network analysis based on multiple microbial domains underlined the topological features in the guts of Caprinae animals, and these findings provide the foundation necessary to understand the mechanism of cross talk between the bacterial microbiota and mycobiota.

**Conclusion.** In summary, we comprehensively investigated the diversity and composition of the gut mycobiota in Caprinae animals from six provinces in China. Concurrently, a co-occurrence network of bacteria and fungi was constructed to explore the cross talk

between the gut bacterial microbiota and mycobiota, as well as their contributions to gut homeostasis and host health. Geographical factors were some of the most important factors affecting the gut fungal composition in Caprinae animals. A better understanding of the gut mycobiota in Caprinae animals will facilitate the development of novel approaches to manage and manipulate gut fungal communities to achieve better animal health and productivity.

## MATERIALS AND METHODS

**Experimental design and sample collection.** The feces of four Caprinae species (*Capra hircus*, $n = 18$; *Ovis aries*, $n = 9$; *Pantholops hodgsonii*, $n = 2$; *Procapra picticaudata*, $n = 1$) from six provinces of China (Anhui, Jilin, Guangxi, Shandong, Shanxi, and Tibet) were collected (Table S1; Fig. S1A). The samples were from healthy adult individuals fed with plants commonly found in the area. To minimize contamination, the animals were stood naturally during sampling. One staff member calmed the animals and another wore disposable sterile PE gloves and placed his finger into the animal's rectum to a distance of 5 cm to collect the feces. For inaccessible animals (e.g., *P. hodgsonii*, also known as Tibetan antelopes), the tracking method was adopted. We collected fecal samples from the central part of the feces immediately after defecation to avoid contact with the outside environment. Detailed information on the animals was provided by local breeders or identified by professionals. The stool samples were immediately transferred to sterile containers for homogenization and stored separately in DNase- and RNase-free centrifuge tubes. All fresh stool samples were frozen in liquid nitrogen and mailed to the laboratory on dry ice within 24 h. Upon arrival, the samples were immediately stored in a −80°C refrigerator for further experiments. The animal experiments were approved by the Qingdao Agriculture University Ethics Committee.

**DNA extraction and sequencing.** Total DNA was extracted from each fecal sample (∼200 mg per sample) using the TIANGEN Magnetic Soil and Stool DNA kit according to the manufacturer's instructions. DNA integrity was determined via 1% agarose gel electrophoresis, and the genomic DNA concentration was determined using a Qubit DNA assay kit with a Qubit 3.0 Flurometer (Invitrogen). The DNA was used for Illumina sequencing with the primers ITS3F: 5'-GCATCGATGAAGAACGCAGC-3' and ITS4R: 5'-TCCTCCGCTTATTGATATGC-3', targeting the ITS-2 region. PCR products were subjected to 250-bp paired-end sequencing on an Illumina NoveSeq 6000.

**Bioinformatics.** Demultiplexed reads were processed via the QIIME2 (v2022.2.0) pipeline using the various built-in plugins (38). The primers were removed from the reads using cutadapt, and the ITS-2 of the fungal reads was extracted from the reads using itsxpress (v 1.8.0) (39). The quality control of reads from each run was performed via the DADA2 algorithm, resulting in the subsequent ASV tables and representative sequences (40). A phylogenetic tree based on representative sequences was constructed by employing the align-to-tree-mafft-fasttree pipeline in phylogeny plugins. This pipeline creates sequence alignments using MAFFT, and any alignment with uninformative or ambiguously phylogenetic information is removed. The resulting masked alignment is used to infer a phylogenetic tree and subsequently rooted at its midpoint. The representative sequences were classified using the feature-classifier plugin from QIIME2, with the classify-sklearn method. The UNITE database (v8.3) (41) was used to train a naive Bayes classifier and to assign fungal taxonomies in QIIME2. Finally, only ASVs with taxonomies assigned to fungal phyla (1,848 ASVs; Table S3) were kept.

**Statistical analyses.** Rarefaction curves based on the observed index were generated using R (version 4.1.1) with the function "estimateR" in the package *vegan* (version 2.5–7). Before statistical analyses, the feature table was transformed into a table of relative abundance of ASVs (Table S2). The Wilcoxon rank sum test was used to compare the differences in taxa between groups. The $\alpha$-diversity was calculated in terms of observed and Shannon indices based on the *vegan* package. The PCoA was based on the Bray-Curtis distance and PERMANOVA with a permutation of 999. The ASV profile was obtained by filtering the ASVs of samples with relative abundances less than 0.01% and 10%. Upset diagrams were visualized in R with the package upsetR (version 1.4.0). Linear discriminant analysis (LDA) of the effect size (LEfSe) (42) was executed to identify the significant taxa that most likely explained the differences between groups, with a threshold LDA score of 2.0. Bacterial and fungal co-occurrence network analysis and visualization was performed by employing the Spearman method (rho cutoff = 0.5, $P < 0.01$) with packages psych (version 2.1.9) and igraph (version 1.2.6) in R and cytoscape software (version 3.7.2), and the bacterial abundance data were taken from the previous feces metagenomic shotgun sequencing data set (20) (Table S7 and S8).

**Data availability.** The raw amplicon sequencing data acquired in this study were deposited at the NCBI under the accession code PRJNA852851. We declare that all other data supporting the findings of this study are available in the paper and supplemental materials or from the corresponding author upon request.

## SUPPLEMENTAL MATERIAL

Supplemental material is available online only.

**SUPPLEMENTAL FILE 1**, XLSX file, 1.7 MB.

**SUPPLEMENTAL FILE 2**, PDF file, 3.6 MB.

## ACKNOWLEDGMENT

This work was supported by the Research Foundation for Distinguished Scholars of Qingdao Agricultural University (665-1120044 and 665-1120046).

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
