## [Reviewer comments · Microbiology Spectrum]

Microbiology Spectrum

Description of gut mycobiota composition and diversity of Caprinae animals

Qing-Bo Lv, Jin-Xin Meng, He Ma, Rui Liu, Ya Qin, Yi-Feng Qin, Hong-Li Geng, Hongbo Ni, and Xiao-Xuan Zhang

Corresponding Author(s): Xiao-Xuan Zhang, Qingdao Agricultural University

Review Timeline:

Submission Date:	July 1, 2022
Editorial Decision:	October 17, 2022
Revision Received:	December 3, 2022
Editorial Decision:	December 5, 2022
Revision Received:	December 16, 2022
Accepted:	December 20, 2022

Editor: Noha Youssef

Reviewer(s): Disclosure of reviewer identity is with reference to reviewer comments included in decision letter(s). The following individuals involved in review of your submission have agreed to reveal their identity: Xiaofan Wang (Reviewer #2)

Transaction Report:

DOI: <https://doi.org/10.1128/spectrum.02424-22>

October 17, 2022

Prof. Xiao-Xuan Zhang
Qingdao Agricultural University
Qingdao, Shandong
China

Re: Spectrum02424-22 (Description of gut mycobiota composition and diversity of Caprinae)

Dear Prof. Xiao-Xuan Zhang:

Thank you for submitting your manuscript to Microbiology Spectrum. Your work has been evaluated by two experts and their comments are appended to this email. When submitting the revised version of your paper, please provide (1) point-by-point responses to the issues raised by the reviewers as file type "Response to Reviewers," not in your cover letter, and (2) a PDF file that indicates the changes from the original submission (by highlighting or underlining the changes) as file type "Marked Up Manuscript - For Review Only". Please use this link to submit your revised manuscript - we strongly recommend that you submit your paper within the next 60 days or reach out to me. Detailed instructions on submitting your revised paper are below.

Link Not Available

Sincerely,

Noha Youssef

Journals Department
Reviewer comments:

Reviewer #1 (Comments for the Author):

In this study, Lv et al. examined the gut mycobiota of Caprinae. This paper investigated the diversity and composition of intestinal fungal communities of sheep family from different locations, and discussed the interaction between gut microbiota and mycobiota. Overall, the topics and results are meaningful and important for the animal husbandry. However, the results are limited by sample size to some extent. Some analysis and writing needs further improvement.

L84. What additives include?

L97-98. Please indicate the manufacturer of the kit used.

L106-108. Check the versions of cutadapt and DADA2

L142. Please unify the abbreviations of region names in Figure1.

- L140. Please provide the rarefaction curve of the relation between the number of ASVs and the number of samples.
- L163. Is *Debaryomyces* a special fungus? Discuss it in the discussion section.
- L166. How is the core microbiota defined? Please provide the criterion of judgment.
- L183. The methods section does not mention the source of the bacterial abundance data. Even if the data has not yet been published, a complete methodological flow should be provided.
- L202-203. The differences in gut mycobiota among caprinae from different regions are not sufficient to prove the influence of geographical factors, so please be cautious in drawing conclusions.
- L207-208. It's just repeating the results.

Reviewer #2 (Comments for the Author):

The Lv et al., comprehensively characterized the composition and diversity of the fungal community in Caprinae in six provinces in China. The efforts allow the understanding of the dominant members of fungi at different locations. Additionally, they have linked the fungal and bacteria to decipher the potential connections between different microbe types. The overall English written is acceptable but requires further grammar edit. The statistical analysis is rigid and can fulfill the results interpretation. I have other insignificant comments below.

1. Line 28. Replace difference with different. Need list the locations briefly in the abstract.
2. Line 37. Better should be used in a comparison situation. Replace the word.
3. Line 44. The gut microbiota already includes bacteria? Please delete one of them.
4. Line 60. Change stage into stages.
5. Line 85. The sentence should stop here. "the animals should stand naturally while sampling, ". Replace comma with period.
6. Line 89. Separate the one long sentence into two. "For inaccessible animals (e.g., *P. hodgsonii*, also known as Tibetan antelopes), the tracking method should be adopted to collect the central part of fecal samples immediately after defecation, a contact with the outside environment should be avoided. "
7. Line 97. Please specify the commercial kit.
8. Line 111. Delete "that was"
9. Line 110-113. Change these sentences into past tense.
10. Line 145. Change discover into discovery.
11. Line 163. Rephrase this sentence "It is noteworthy that *Debaryomyces* was the higher abundance genus". "Higher" should be used in a comparison situation.
12. Line 168. Please clarify the way of proportion being calculated.
13. Line 177. Change genus into genera.
14. Line 196. Please rephrase the statement "no studies have been performed for exp...". Better to change it to "few studies".

Staff Comments:

Preparing Revision Guidelines

Please return the manuscript within 60 days; if you cannot complete the modification within this time period, please contact me. If you do not wish to modify the manuscript and prefer to submit it to another journal, please notify me of your decision immediately so that the manuscript may be formally withdrawn from consideration by Microbiology Spectrum.

If your manuscript is accepted for publication, you will be contacted separately about payment when the proofs are issued; please follow the instructions in that e-mail. Arrangements for payment must be made before your article is published. For a

complete list of **Publication Fees**, including supplemental material costs, please visit our website.

29 November, 2022

Prof. Noha Youssef
Editor-in-Chief
Microbiology Spectrum

Dear Prof. Noha Youssef,

Re: Revised Manuscript ID Spectrum02424-22

On behalf of all co-authors, I would like to thank you and the two reviewers very much for their careful review and constructive suggestions with regard to our manuscript (MS) ID Spectrum02424-22. These comments are all valuable and very helpful for revising and improving our paper, as well as the importance guiding significance to our MS. We have studied comments carefully and have made correction which we hope meet with approval.

Responses to the comments and suggestions of Reviewer #1:

Reviewer #1 (Comments for the Author):

In this study, Lv et al. examined the gut mycobiota of Caprinae. This paper investigated the diversity and composition of intestinal fungal communities of sheep family from different locations, and discussed the interaction between gut microbiota and mycobiota. Overall, the topics and results are meaningful and important for the animal husbandry. However, the results are limited by sample size to some extent. Some analysis and writing needs further improvement.

Point 1. L84. What additives include?

Responses: We thank **Reviewer #1** very much for his/her constructive comments and suggestions on our MS. This was an incorrect description and we removed it.

Point 2. L97-98. Please indicate the manufacturer of the kit used.

Responses: We thank **Reviewer #1** very much for his/her constructive comments and suggestions on our MS. We added brand and manufacturer of kit in the methods section.

Point 3. L106-108. Check the versions of cutadapt and DADA2

Responses: We thank **Reviewer #1** very much for his/her constructive comments and suggestions on our MS. The cutadapt and DADA2 are plugins packaged into Qiime2, so the

versions are consistent with Qiime2. However, in order not to create ambiguity, we do not list these duplicate version numbers separately.

Point 4. L142. Please unify the abbreviations of region names in Figure1.

Responses: We thank **Reviewer #1** very much for his/her constructive comments and suggestions on our MS. We corrected the Figure1.

Point 5. L140. Please provide the rarefaction curve of the relation between the number of ASVs and the number of samples.

Responses: We thank **Reviewer #1** very much for his/her constructive comments and suggestions on our MS. We supplemented this analysis and provided the visual results to the reviewers. The analysis showed that the curve did not flatten out at the end, indicating that the sample size of this study limited our findings. We will further supplement these data in future studies.

Point 6. L163. Is *Debaryomyces* a special fungus? Discuss it in the discussion section.

Responses: We thank **Reviewer #1** very much for his/her constructive comments and suggestions on our MS. *Debaryomyces* is an ascomycetous yeast commonly found in soil, water and plants. We added a description of this fungus genus in the discussion section.

Point 7. L166. How is the core microbiota defined? Please provide the criterion of judgment.

Responses: We thank **Reviewer #1** very much for his/her constructive comments and suggestions on our MS. The core ASVs is defined as ASVs that existed in at least 50% of the samples of each province.

Point 8. L183. The methods section does not mention the source of the bacterial abundance data. Even if the data has not yet been published, a complete methodological flow should be provided.

Responses: We thank **Reviewer #1** very much for his/her constructive comments and suggestions on our MS. These methods have been published in previous articles to which we have added citations. Based on metagenomic shotgun sequencing, we reconstructed the bacterial and fungal genomes of these stool samples. Raw data mapping to the constructed genome catalog quantified the relative abundance of these bacteria and archaea.

Point 9. L202-203. The differences in gut mycobiota among Caprinae from different regions are not sufficient to prove the influence of geographical factors, so please be cautious in drawing conclusions.

Responses: We thank **Reviewer #1** very much for his/her constructive comments and suggestions on our MS. We recast these results and use a more cautious tone of conclusion.

Point 10. L207-208. It's just repeating the results.

Responses: We thank **Reviewer #1** very much for his/her constructive comments and suggestions on our MS. We deleted this part of the duplicate content.

Responses to the comments and suggestions of Reviewer #2:

Reviewer #2 (Comments for the Author):

The Lv et al., comprehensively characterized the composition and diversity of the fungal community in Caprinae in six provinces in China. The efforts allow the understanding of the dominant members of fungi at different locations. Additionally, they have linked the fungal and bacteria to decipher the potential connections between different microbe types. The overall English written is acceptable but requires further grammar edit. The statistical analysis is rigid and can fulfill the results interpretation. I have other insignificant comments below.

Point 1. Line 28. Replace difference with different. Need list the locations briefly in the abstract.

Responses: We thank **Reviewer #2** very much for his/her constructive comments and suggestions on our MS. We have supplemented the content according to the reviewer's comments.

Point 2. Line 37. Better should be used in a comparison situation. Replace the word.

Responses: We thank **Reviewer #2** very much for his/her constructive comments and suggestions on our MS. We have supplemented the content according to the reviewer's comments.

Point 3. Line 44. The gut microbiota already includes bacteria? Please delete one of them.

Responses: We thank **Reviewer #2** very much for his/her constructive comments and suggestions on our MS. We have supplemented the content according to the reviewer's comments.

Point 4. Line 60. Change stage into stages.

Responses: We thank **Reviewer #2** very much for his/her constructive comments and suggestions on our MS. We have supplemented the content according to the reviewer's comments.

Point 5. Line 85. The sentence should stop here. "The animals should stand naturally while sampling, ". Replace comma with period.

Responses: We thank **Reviewer #2** very much for his/her constructive comments and suggestions on our MS. We have supplemented the content according to the reviewer's comments.

Point 6. Line 89. Separate the one long sentence into two. "For inaccessible animals (e.g., *P. hodgsonii*, also known as Tibetan antelopes), the tracking method should be adopted to collect the central part of fecal samples immediately after defecation, a contact with the outside environment should be avoided. "

Responses: We thank **Reviewer #2** very much for his/her constructive comments and suggestions on our MS. We have supplemented the content according to the reviewer's comments.

Point 7. Line 97. Please specify the commercial kit.

Responses: We thank **Reviewer #2** very much for his/her constructive comments and suggestions on our MS. We added brand and manufacturer of kit in the methods section.

Point 8. Line 111. Delete "that was"

Responses: We thank **Reviewer #2** very much for his/her constructive comments and suggestions on our MS. We have supplemented the content according to the reviewer's comments.

Point 9. Line 110-113. Change these sentences into past tense.

Responses: We thank **Reviewer #2** very much for his/her constructive comments and suggestions on our MS. We have supplemented the content according to the reviewer's comments.

Point 10. Line 145. Change discover into discovery.

Responses: We thank **Reviewer #2** very much for his/her constructive comments and suggestions on our MS. We have supplemented the content according to the reviewer's comments.

Point 11. Line 163. Rephrase this sentence "It is noteworthy that *Debaryomyces* was the higher abundance genus". "Higher" should be used in a comparison situation.

Responses: We thank **Reviewer #2** very much for his/her constructive comments and suggestions on our MS. We have supplemented the content according to the reviewer's comments.

Point 12. Line 168. Please clarify the way of proportion being calculated.

Responses: We thank **Reviewer #2** very much for his/her constructive comments and suggestions on our MS. This refers to the proportion of specific ASVs to the total number of ASVs. We added a note to the manuscript.

Point 13. Line 177. Change genus into genera.

Responses: We thank **Reviewer #2** very much for his/her constructive comments and suggestions on our MS. We have supplemented the content according to the reviewer's comments.

Point 14. Line 196. Please rephrase the statement "no studies have been performed for exp...". Better to change it to "few studies".

Responses: We thank **Reviewer #2** very much for his/her constructive comments and

suggestions on our MS. We have supplemented the content according to the reviewer's comments.

December 5, 2022

Prof. Xiao-Xuan Zhang
Qingdao Agricultural University
Qingdao, Shandong
China

Re: Spectrum02424-22R1 (Description of gut mycobiota composition and diversity of Caprinae)

Dear Prof. Xiao-Xuan Zhang:

Thank you for addressing the reviewers comments. At this level, the manuscript still requires English language editing.

Thank you for submitting your manuscript to Microbiology Spectrum. As you will see your paper is very close to acceptance. Please modify the manuscript along the lines I have recommended. As these revisions are quite minor, I expect that you should be able to turn in the revised paper in less than 30 days, if not sooner. If your manuscript was reviewed, you will find the reviewers' comments below.

When submitting the revised version of your paper, please provide (1) point-by-point responses to the issues raised by the reviewers as file type "Response to Reviewers," not in your cover letter, and (2) a PDF file that indicates the changes from the original submission (by highlighting or underlining the changes) as file type "Marked Up Manuscript - For Review Only". Please use this link to submit your revised manuscript. Detailed instructions on submitting your revised paper are below.

Link Not Available

Sincerely,

Noha Youssef

Reviewer comments:

Preparing Revision Guidelines

Please return the manuscript within 60 days; if you cannot complete the modification within this time period, please contact me. If you do not wish to modify the manuscript and prefer to submit it to another journal, please notify me of your decision immediately so that the manuscript may be formally withdrawn from consideration by Microbiology Spectrum.

We certify that the following article

Description of gut mycobiota composition and diversity of Caprinae

Xiao-Xuan Zhang

has undergone English language editing by MDPI. The text has been checked for correct use of grammar and common technical terms, and edited to a level suitable for reporting research in a scholarly journal.

MDPI uses experienced, native English speaking editors. Full details of the editing service can be found at

► <https://www.mdpi.com/authors/english>.

Basel, Switzerland
December 2022

December 20, 2022

Prof. Xiao-Xuan Zhang
Qingdao Agricultural University
Qingdao, Shandong
China

Re: Spectrum02424-22R2 (Description of gut mycobiota composition and diversity of Caprinae animals)

Dear Prof. Xiao-Xuan Zhang:

Thank you for your modification to the manuscript.

Your manuscript has been accepted, and I am forwarding it to the ASM Journals Department for publication. You will be notified when your proofs are ready to be viewed.

Sincerely,

Noha Youssef
Editor, Microbiology Spectrum
